# Single-Sample Networks Reveal Intra-Cytoband Co-Expression Hotspots in Breast Cancer Subtypes

**DOI:** 10.3390/ijms252212163

**Published:** 2024-11-13

**Authors:** Richard Ponce-Cusi, Patricio López-Sánchez, Vinicius Maracaja-Coutinho, Jesús Espinal-Enríquez

**Affiliations:** 1Advanced Center for Chronic Diseases—ACCDiS, Facultad de Ciencias Químicas y Farmacéuticas, Universidad de Chile, Santiago 8330015, Chile; bio.richard.ponce@gmail.com; 2Escuela Profesional de Medicina, Facultad de Ciencias de la Salud, Universidad Nacional de Moquegua, Moquegua 180101, Peru; 3Computational Genomics Division, National Institute of Genomic Medicine, Mexico City 14610, Mexico; patrick.losa90@gmail.com; 4Unidad de Genómica Avanzada—UGA, Facultad de Ciencias Químicas y Farmacéuticas, Universidad de Chile, Santiago 8330015, Chile

**Keywords:** single-sample networks, co-expression networks, breast cancer networks, intra-chromosomal hotspots, intra-cytoband co-expression

## Abstract

Breast cancer is a heterogeneous disease comprising various subtypes with distinct molecular characteristics, clinical outcomes, and therapeutic responses. This heterogeneity evidences significant challenges for diagnosis, prognosis, and treatment. Traditional genomic co-expression network analyses often overlook individual-specific interactions critical for personalized medicine. In this study, we employed single-sample gene co-expression network analysis to investigate the structural and functional genomic alterations across breast cancer subtypes (Luminal A, Luminal B, Her2-enriched, and Basal-like) and compared them with normal breast tissue. We utilized RNA-Seq gene expression data to infer gene co-expression networks. The LIONESS algorithm allowed us to construct individual networks for each patient, capturing unique co-expression patterns. We focused on the top 10,000 gene interactions to ensure consistency and robustness in our analysis. Network metrics were calculated to characterize the topological properties of both aggregated and single-sample networks. Our findings reveal significant fragmentation in the co-expression networks of breast cancer subtypes, marked by a change from interchromosomal (TRANS) to intrachromosomal (CIS) interactions. This transition indicates disrupted long-range genomic communication, leading to localized genomic regulation and increased genomic instability. Single-sample analyses confirmed that these patterns are consistent at the individual level, highlighting the molecular heterogeneity of breast cancer. Despite these pronounced alterations, the proportion of CIS interactions did not significantly correlate with patient survival outcomes across subtypes, suggesting limited prognostic value. Furthermore, we identified high-degree genes and critical cytobands specific to each subtype, providing insights into subtype-specific regulatory networks and potential therapeutic targets. These genes play pivotal roles in oncogenic processes and may represent important keys for targeted interventions. The application of single-sample co-expression network analysis proves to be a powerful tool for uncovering individual-specific genomic interactions.

## 1. Introduction

Breastcancer remains the most prevalent malignancy among women worldwide and a leading cause of cancer-related mortality [1]. Its complexity arises from a notable molecular heterogeneity that manifests both between patients and within individual tumors, which poses significant challenges for diagnosis, prognosis, and treatment, due to breast cancer encompassing a range of diseases with distinct genetic and epigenetic profiles [2]. In general terms, breast cancer is classified into molecular subtypes, such as Luminal A, Luminal B, HER2-enriched and Basal-like, each one characterized by specific gene expression patterns, clinical outcomes and responses to treatment [3,4]. Despite advances in targeted therapies and personalized medicine, areas such as therapeutic resistance and variable patient outcomes persist without complete resolution, emphasizing the need for a deeper understanding of the molecular basis driving these subtypes.

In this context, gene co-expression networks (GCNs) have emerged as powerful tools in systems biology for unraveling the intricate relationships between genes based on expression data [5]. By modeling genes as nodes and their co-expression relationships as edges, GCNs allow for the structural analysis of gene–gene relationships, the discovery of functionally connected gene modules and key regulatory elements involved in cellular processes and pathologies [6,7]. Integrating co-expression network analysis with clinical data enhances our ability to understand disease mechanisms, predict therapeutic responses, and guide precision medicine initiatives.

The dynamic nature of cancer is further compounded by genetic instability and epigenetic alterations, contributing to the evolution of tumors and their ability to evade standard treatments [8]. Traditional approaches that consider average behaviors across patient populations often fail to capture the intricacies of individual tumors. Therefore, there is a critical need for analytical frameworks that can elucidate the complex gene regulatory networks at the individual level, facilitating the identification of patient-specific biomarkers and therapeutic targets [9,10].

In that sense, advancements in network biology have led to the development of single-sample gene co-expression network methodologies [11,12,13,14]. This approach focuses on individual variations in gene expression, allowing for the construction of personalized networks that capture the unique co-expression patterns from a specific patient tumor sample. Such personalized network analyses are particularly relevant in heterogeneous diseases like breast cancer, as they provide insights into patient-specific pathologies and uncover potential targets that might be overlooked in population-level analyses [15].

In this study, we aim to examine the structure and dynamics of single-sample gene co-expression networks in breast cancer subtypes. By employing a LIONESS [12,16], a robust computational framework to construct and analyze these networks, we seek to understand the complex genomic interactions at the individual level. Our objectives are to (1) identify distinctive co-expression patterns associated with each molecular subtype, (2) explore the relationship between network topology and clinical characteristics, and (3) assess the potential of single-sample network features as predictive markers for patient outcomes.

## 2. Results

### 2.1. Aggregated Networks Disruption Across Breast Cancer Subtypes

To establish a comprehensive understanding of the global structural properties of gene co-expression in breast cancer, the analysis was initially performed on aggregated networks that combine data from multiple samples within each breast cancer molecular subtype and healthy tissue. The generated co-expression networks analysis provides an integrated view of genomic interaction dynamics within the different sample types analyzed, as depicted in Figure 1. In our analysis, healthy breast tissue exhibited a large, interconnected component dominated by interchromosomal interactions, indicating a highly organized genomic architecture. In contrast, the breast cancer subtypes displayed fragmented structures, characterized by intrachromosomal interactions, highlighting compromised genomic integrity, which likely implies a breakdown in long-range genomic regulation. The network patterns underscore the genomic heterogeneity across subtypes, with Luminal A featuring intrachromosomal interactions reflecting localized genomic regulation. However, Luminal B demonstrated higher fragmentation and more intrachromosomal interactions than Luminal A. In contrast Her2 presented moderate fragmentation, while the Basal subtype exhibited the highest degree of fragmentation, with generalized intrachromosomal interactions.

### 2.2. Intrachromosomal and Intracytoband Interaction Dynamics in Aggregated Networks

The fragmentation observed in Figure 1 led to a deeper analysis of the aggregated networks, as depicted in the upper right part of Figure 1. We focused on the specific interaction patterns between intrachromosomal (CIS) and interchromosomal (TRANS) interactions, as well as their distribution across chromosomal cytobands. In healthy breast tissue, TRANS interactions are more prevalent than CIS, reflecting a genomic landscape marked by long-range communication. In cancer subtypes, however, this balance changes markedly, with a pronounced increase in CIS interactions (red bars).

This transition indicates a change toward localized genomic interactions, contributing to the broader network disintegration previously presented, where the loss of long-range genomic interactions may compromise normal regulatory communication. Additionally, the analysis of intra- and inter-cytoband interactions reinforces this trend. Healthy individuals displayed fewer intra- and inter-cytoband interactions, with a higher occurrence of interchromosomal interactions, indicative of a more integrated genomic structure (Figure 2).

In contrast, cancer subtypes exhibit a notable rise in both intra- and inter-cytoband interactions, with a reduction in interchromosomal interactions. This transition to regulatory interactions within specific chromosomal regions highlights the loss of global genomic coordination, suggesting localized instability and fragmented network architecture in cancer.

### 2.3. Breast Cancer Heterogeneity in Genomic Interactions According to Co-Expression Aggregated Networks

For a deeper analysis of the aggregated networks, we also identified genomic regions with the highest interaction frequencies by observing those chromosomes and cytobands with more interactions, as shown in Table 1 and Table 2.

In healthy tissue, intrachromosomal interactions are primarily concentrated in chromosomes 1, 19, and 2. In contrast, cancer subtypes exhibit distinct interaction patterns. Luminal A and Luminal B showed high interaction frequencies in chromosomes 17, 11, and 8, while Her2 primarily involved the chromosome 17. Basal, the most aggressive subtype, showed higher interactions in chromosomes 1, 19, 9 and 10 (Table 1). When examining intracytoband interactions, healthy tissue presented the highest frequencies in cytobands 6p21.32, 8p11.23, and 8q24.3. However, cancer subtypes exhibit a shift towards specific cytobands. Luminal A and Luminal B cluster predominantly in 11q13.1, 8q24.3, and 17q11.2, while Her2 subtypes were enriched in 17q11.2 and 8q24.3. Basal subtype showed significant intracytoband interactions in 6p21.1 and 8q24.3 (Table 2).

In addition to identifying the frequency of interactions, we assessed the proportion of these interactions relative to the total number of genes in each chromosome or cytoband. Healthy tissue demonstrated lower proportions of interactions in chromosomes and cytobands, reflecting its more distributed regulatory activity. In contrast, breast cancer subtypes, exhibited significantly higher proportions of interactions in specific regions, such as chromosome 17 and cytoband 17q11.2.

### 2.4. Interaction Patterns in Single-Sample Co-Expression Networks

Following the analysis of aggregated networks, we applied a detailed evaluation of CIS/TRANS and intra/inter cytoband interaction patterns in single-sample networks, allowing us to capture unique co-expression profiles for each patient. In healthy tissue, there is a predominance of TRANS interactions over CIS, reflecting a coordinated genomic system characterized by extensive cross-chromosomal communication, as observed in the aggregated network.

Conversely, breast cancer subtypes, particularly Luminal A, Luminal B, and Basal, exhibit an increase in CIS interactions, indicating a change towards localized genomic regulation and disrupted long-range coordination. This pattern, less pronounced in single samples than in aggregated networks, still highlights a key regulatory disruption common across patients (Figure 2A).

Despite the increased frequency of CIS interactions, TRANS interactions remain slightly more frequent than CIS in cancer subtypes, suggesting that long-range genomic communication is not entirely lost but significantly compromised. The persistence of these disruptions across individual tumor samples emphasizes the unique co-expression patterns specific to each tumor, contributing to the molecular complexity of breast cancer. Furthermore, Figure 2B expands on this by illustrating the distribution of the intra- and intercytoband, as well as interchromosomal interactions.

In healthy tissue, intracytoband interactions are sparse, while intercytoband interactions occur with moderate frequency, indicating minimal network fragmentation. In contrast, cancer subtypes, particularly Luminal A, Luminal B, and Basal, display an increase in intra- and intercytoband interactions, suggesting a reorganization of co-expression networks that favors chromosomal region-specific interactions, highlighting the increased genomic instability and network fragmentation observed in cancer. This trend complements the broader network disintegration seen in the aggregated networks, confirming the general transition toward localized communication control in cancer.

### 2.5. Chromosomal and Cytoband Interaction Patterns in Single-Sample Networks

The analysis of single-sample co-expression networks provided a detailed view of the genomic regions with the highest proportions of intrachromosomal and intracytoband interactions in breast cancer subtypes and normal tissue. Unlike aggregated networks, where proportions were evaluated for a single network, here, we quantified the number of samples in which each chromosome and cytoband exhibited the highest proportion of interactions. As shown in Table 1, chromosome 1 is identified in 113 normal samples as having the highest proportion of intrachromosomal interactions. In Luminal A samples, chromosomes 17, 11, and 1 are frequently associated with higher proportions of intrachromosomal interactions across numerous samples, pointing to key regulatory regions in this subtype. Similarly, Luminal B presented frequent interactions in chromosomes 11 and 17, indicating shared genomic vulnerabilities with Luminal A. Her2 samples predominantly presented interactions in chromosome 17, while Basal networks exhibited prominent interactions in chromosomes 1 and 19. In the same way, Table 2 illustrates the distribution of cytoband interactions, where normal tissue showed the most frequent occurrences in cytobands 3p21.31, 6p21.32, and 17q11.2 across multiple samples. In Luminal A, 11q13.1, 8q24.3, and 19p13.3 emerged as recurrent cytobands, while Luminal B displayed significant activity in 8q24.3 and 17q11.2, underscoring shared features with Luminal A. Her2 subtypes are more enriched on 17q11.2 and 17q25.3, whereas Basal samples prominently featured cytobands 8q24.3 and 6p21.1. These findings highlight the molecular heterogeneity and chromosomal region-specific disruptions present across breast cancer subtypes.

Additionally, similar to the findings in aggregated networks, healthy tissue showed lower interaction proportions across chromosomes and cytobands, reflecting more distributed genomic interaction. Conversely, cancer subtypes exhibited significantly higher proportions of interactions in regions such as chromosome 17 and cytobands 8q24.3, and 17q11.2, among others, underscoring concentrated regulatory disruptions. This trend aligns with observations from aggregated networks.

### 2.6. Single-Sample Network Analysis Reveals Key Topological Differences in Breast Cancer Subtypes

Following the detailed examination of CIS/TRANS and cytoband interaction patterns in single-sample networks, we extended the analysis to key network metrics to further elucidate the structural properties of these co-expression networks. Network metrics can provide relevant information on topological differences between healthy and cancerous tissues and inform how they are organized and maintained within individual patients.

In this respect, Figure 3 evidences differences in single-network metrics between healthy and cancer samples. In healthy samples, the clustering coefficient is moderate, reflecting a balance of local cohesion, whereas Luminal A, Luminal B, and Basal subtypes exhibited higher clustering, indicating denser local interactions. Her2, by contrast, presented a lower clustering coefficient, suggesting less local connectivity. Modularity, a measure of the network’s community structure, is moderate in healthy samples but significantly higher in Luminal A, Luminal B, and Basal, suggesting more defined substructures in these cancer subtypes. Closeness centrality, which reflects node accessibility, is higher in healthy networks, revealing a well-balanced distribution of interactions. However, in all breast cancer subtypes it is lower, pointing to less centralized networks that may influence dysregulated regulatory dynamics in cancer samples. Degree, representing the number of connections per node, is higher in healthy samples, implying more robust gene connectivity, in contrast, all subtypes showed fewer connections, indicating a potential loss of regulatory complexity. Healthy networks also demonstrated higher global efficiency, with all cancer subtypes showing reduced efficiency, suggesting less effective gene interactions. Finally, network density is notably higher in healthy samples, highlighting a more connected network, whereas the cancer subtype exhibited significantly lower density, reflecting a fragmented and less cohesive gene interaction network.

### 2.7. Comparative Co-Expression Patterns Between Aggregated and Single-Sample Networks

Understanding the heterogeneity within cancer subtypes is key to uncovering the complexity of tumor biology. By comparing aggregated networks with their single-sample counterparts, we can assess how much individual patient networks diverge from the generalized patterns. The results depicted in Figure 4 compare aggregated networks and their respective single-sample counterparts for breast cancer subtypes and normal tissue, using the Jaccard index. This analysis highlights the top 10,000 interactions (ranked by absolute mutual information values) in both network types. Normal tissue showed a lower similarity between aggregated and single-sample networks, indicating higher variability in individual network interactions compared to the aggregated network. This suggests that individual samples in normal tissue exhibit more heterogeneity in their co-expression patterns. On the other hand, molecular subtypes (Luminal A, Luminal B, Her2, and Basal) displayed higher Jaccard indices, reflecting greater similarity between their individual and aggregated networks. However, despite this increased similarity, there is still notable heterogeneity in the single-sample networks of breast cancer subtypes, as seen by the gradual decline in similarity scores, which may contribute to the complexity of its biology.

### 2.8. The Proportion of CIS Interactions Occurring in Single-Sample Co-Expression Networks and Survival Outcomes

To investigate the prognostic relevance of intrachromosomal (CIS) interactions from single-sample co-expression networks in breast cancer, we evaluated the relationship between CIS interaction proportions and overall survival. This analysis was conducted across multiple molecular subtypes, providing insight into whether the CIS interaction proportion impacts patient outcomes. Kaplan–Meier survival curves were stratified by high and low CIS interaction proportions and revealed no significant differences in overall survival across all subtypes, as indicated by log-rank tests (Luminal A: *p* = 0.83; Luminal B: *p* = 0.79; Her2: *p* = 0.68; Basal: *p* = 0.72). In Luminal A and Luminal B subtypes, the curves for high and low CIS groups overlapped significantly, suggesting that intrachromosomal stability does not influence overall survival in these patient groups. Similarly, the Her2 subtype showed no significant separation between the survival curves. Although the Basal subtype is characterized by higher clinical aggressiveness, the lack of significant differences between high and low CIS groups suggests that the proportion of CIS interactions is not a major determinant of overall survival in this subtype.

In addition, a 5-year survival analysis further confirmed the absence of a significant association between CIS link proportions and survival (Figure 5), with *p*-values remaining non-significant across all subtypes (Luminal A: *p* = 0.8; Luminal B: *p* = 0.74; Her2: *p* = 0.47; Basal: *p* = 0.18). In the Luminal subtypes, this evaluation of survival outcomes continued to show substantial overlap between high and low CIS groups, reinforcing the conclusion that intrachromosomal proportion is not a critical factor in patient prognosis. While the Her2 subtype displayed a slight divergence in survival curves, this did not reach statistical significance, and the Basal subtype, despite showing the lowest *p*-value, still did not present a significant difference. Collectively, these findings suggest that the proportion of CIS interactions is not a predictor of breast cancer survival.

### 2.9. High-Degree Gene Distribution and Cytoband Localization in Single-Sample Networks

High-degree genes play a pivotal role in shaping the co-expression landscape across different breast cancer subtypes. Through single-sample co-expression network analysis, the recurrence of specific high-degree genes, along with their localization in critical genomic regions, provides insight into molecular heterogeneity. In this regard, Figure 6 and Figure 7 provide complementary analyses of high-degree genes within single-sample co-expression networks for each breast cancer subtype. This analysis highlights the top 10,000 interactions (ranked by absolute mutual information values), emphasizing their critical roles in the largest network components of each sample. Figure 6 underscores the recurrence of specific high-degree genes within each subtype: C1QBP and RPS3A in Luminal A, USP31 and RRM1 in Luminal B, ANKRD30A and PUF60 in Her2, and PSMD8 and YME1L1 in Basal. In Figure 7, the localization of these genes across cytobands further elucidates the genomic regions where these regulatory disruptions occur. Luminal A and Luminal B share key cytobands, such as 11q13.1 and 17q12.1, while 8q24.3 emerges as a recurrent region in Luminal B and Her2 subtypes. Basal subtypes emphasize cytobands such as 6p21.1 and 19q13.2, indicating key areas driving their aggressive nature.

## 3. Discussion

The findings from our co-expression network analysis highlight significant differences in network topology between healthy breast tissue and breast cancer subtypes, reflecting underlying molecular heterogeneity and genomic instability. In healthy tissue, the presence of a large giant component with extensive interchromosomal interactions suggests a highly coordinated genomic architecture essential for maintaining normal cellular function. In contrast, the fragmentation and prevalence of intrachromosomal interactions observed in cancer networks suggest disrupted long-range genomic communication. This pattern aligns with previous studies that have documented a loss of TRANS-co-expression in cancer tissues, which is thought to contribute to the breakdown of normal regulatory mechanisms and promote localized genomic instability [17,18,19,20].

The change from the predominant TRANS interaction to the CIS and the increase in intra- and intercytoband interactions observed in cancer subtypes (Figure 2) accentuate the idea of disruption of long-term genomic communication and the transition to a more local regulatory framework, which reflects the fundamental decomposition of genomic architecture coordination, and is consistent with recent studies where such fragmentation and increased intrachromosomal interactions are associated with genomic instability and local chromosome rearrangements in cancer [21,22,23]. These localized interactions, especially the increase in intra- and inter cytoband interactions, indicate a transition towards chromosome instability where certain genomic regions become hotspots for mutations and rearrangements, facilitating oncogenic transformation [24,25]. This interruption of long-term interactions can lead to the formation of transcriptionally active domains, which may promote specific cancer gene expression patterns and contribute to the aggressive behavior and heterogeneity observed in breast cancer subtypes [26]. Moreover, the accumulation of these localized interactions could create genomic regions with sustained aberrant expression, enhancing cellular plasticity and adaptability in response to external stresses or therapeutic interventions. This emphasizes the dynamic reorganization of chromosomal regions as a hallmark of cancer progression. The results highlight the importance of understanding the genomic spatial organization in cancer.

Similarly, the results of the single-sample co-expression network analysis of the balance of CIS and TRANS interactions between healthy and cancerous tissues showed that while the initial aggregated network presented a marked increase in CIS interactions within breast cancer subtypes, the strength of the current results lies in the individualized analysis of single-sample co-expression networks. Remarkably, even at the single-sample level, the CIS/TRANS interaction patterns remain consistent with the aggregated network findings, supporting the robustness of these trends across different levels of analysis [17,27,28,29]. This provides compelling evidence that the disruption of long-range genomic communication is not an artifact of network aggregation, but may be considered a fundamental feature of the genomic instability of cancer. This transition toward increased CIS interactions likely reflects localized genomic alterations that contribute to chromosomal rearrangements or focal amplifications, which are common features of cancer. This rearrangement may foster genomic regions that become transcriptionally active or prone to instability, thus favoring tumor heterogeneity and progression by enabling localized oncogenic signaling or mutational hotspots.

Importantly, the individualized nature of this analysis resolves many of the limitations associated with aggregated network studies, where data from multiple samples are combined, potentially masking the unique genomic interactions that occur within individual tumors. In this regard, single-sample networks offer a more precise view of tumor-specific genomic coordination and the localized genomic disruptions characteristic of different types of cancer [12,15,16]. This individual-based approach is crucial, as genomic heterogeneity is a well-established feature of cancer, and aggregated analyses often fail to capture the full extent of variability between tumors [30]. Studies of individual genomic profiles have demonstrated that single-sample analyses are more effective at identifying clinically relevant mutations and genomic alterations that drive tumor behavior, providing a clearer path for personalized therapeutic interventions [14,31]. The increase in CIS interactions and the reconfiguration of co-expression networks toward intra- and inter cytoband interactions supports the notion of increased genomic instability. These changes align with prior research showing that cancer cells often reorganize their chromatin structures to enhance localized genomic interactions, promoting the formation of transcriptional hubs that regulate oncogene expression and contribute to tumor growth [19,21,29]. Furthermore, capturing heterogeneity in coexpression patterns at the individual level would enable the identification of unique genomic alterations that could foster the development of tailored therapeutic strategies. Thus, integrating genomic analyses of individual samples into oncology research may provide deeper insight into underlying genomic instability and help identify potential biomarkers or novel patient-specific therapeutic targets.

In this study, we conducted two types of co-expression network analyses: aggregated networks and single-sample networks. Aggregated networks were constructed by combining gene expression data from multiple samples within each breast cancer subtype and healthy tissue, producing a collective overview of genomic interactions in five networks, one for each breast cancer subtype (Luminal A, Luminal B, Her2, Basal) and one for the normal tissue phenotype. The goal was to identify the most frequent intra-chromosomal and intra-cytoband interactions across these combined networks. In contrast, single-sample networks were built for each patient, focusing on capturing unique genomic interactions at the individual level. Here, we analyzed which chromosomes and cytobands exhibited the highest number of intra-chromosomal and intra-cytoband interactions across multiple individual samples. This dual approach, reflected in the results summarized in Table 1 and Table 2, allowed us to compare broad, population-level disruptions with patient-specific genomic rearrangements, highlighting similarities and distinctions between aggregated and single-sample networks.

Furthermore, Table 1 and Table 2 underscore the importance of understanding the regulatory hotspots that differentiate healthy breast tissue from breast cancer subtypes through the marked prevalence of intrachromosomal and cytoband interactions with both approaches. The intrachromosomal interactions, as shown in these tables, reveal that chromosome 1 plays a critical role in maintaining genomic stability in healthy samples, likely due to its size and its involvement in essential cellular processes, such as metabolism and DNA repair [32]. In contrast, cancerous tissues exhibit distinct chromosomal preferences; in Luminal subtypes, chromosomes 11, 17, and 1 showed greater interaction frequency, likely reflecting how cancer cells reorganize chromatin to enhance localized regulatory interactions [27,28,33,34,35]. This aligns with existing literature that identifies chromosomes 17 and 11 as regions frequently involved in breast cancer progression, primarily due to the presence of oncogenes like ERBB2 (Her2) and CCND1, which are known drivers of cell proliferation and tumor growth through gene amplification [36,37]. The prevalence of chromosome 17 in Her2 and Basal subtypes also correlates with the more aggressive clinical behavior of these cancers, as this region contains key genes involved in DNA repair and cell cycle control, whose dysregulation contributes to rapid tumor development [34,38].

In terms of cytoband interactions, healthy tissue samples demonstrate a strong predominance of interactions in cytoband 6p21.32, highlighting the role of immune-related genes, particularly those associated with the major histocompatibility complex (MHC), in preserving genomic integrity and preventing cancer development [39]. By contrast, the increased interactions within cytoband 11q13.1 in Luminal A and B subtypes are indicative of dysregulation in regions harboring CCND1, a well-known regulator of the cell cycle, which is often amplified in breast cancer and drives abnormal cell proliferation [40,41]. Cytoband 8q24.3, enriched in Luminal B and Her2 subtypes, is another hotspot for oncogene amplification, specifically for MYC, which plays a pivotal role in controlling cell growth and metabolism and is associated with poor prognosis in these subtypes due to its role in driving aggressive tumor behavior [42,43]. The increased presence of interactions in cytobands 17q11.2 and 8q24.3 in Her2 and Basal subtypes underscores the shared genomic vulnerabilities of these subtypes, as both regions are linked to critical oncogenes like ERBB2 and MYC, whose amplification exacerbates genomic instability, promoting resistance to therapy and aggressive tumor progression [44,45]. Additionally, Basal subtypes exhibit notable interactions in cytoband 6p21.1, suggesting that inflammatory and immune-related pathways contribute to the aggressive nature of this subtype, compounding the already complex genomic instability present in these tumors [46,47].

Concerning the structural analysis of the single networks presented in Figure 3, which provides critical insights into the molecular organization of breast cancer subtypes, key differences in network metrics highlighted distinct genomic topology patterns that differentiate healthy tissues from cancer subtypes. The clustering coefficient, which reflects local connectivity in the network, is significantly higher in Luminal A, Luminal B, and Basal subtypes compared to healthy tissues. This increase suggests that cancer subtypes may form denser local interactions, potentially representing tightly coordinated gene modules that drive oncogenic signaling. In this context, higher clustering often indicates that genes within these networks are highly co-expressed, forming clusters of genes that may promote cancer progression through the formation of abnormal regulatory loops that can contribute to maintaining the proliferative advantage of cancer cells by reinforcing the robustness of pathways related to mechanisms such as cell cycle regulation, growth factor signaling, or resistance to apoptosis. Such dense clusters may reflect a change towards tumor-promoting interactions, enabling subtypes like Luminal B or Basal to evade the normal checkpoints that restrict cell growth [17,48,49,50].

Modularity measures the degree of network compartmentalization into distinct gene communities [51]. The significantly higher modularity observed in the Luminal A, Luminal B, and Basal subtypes suggests that these tumors develop specialized genetic interaction networks that could allow tumor cells to adapt to the stress of the tumor microenvironment. By forming well-defined substructures, these modules may facilitate the activation of oncogenic pathways essential to tumor survival and proliferation. Furthermore, this increase in modularity may also indicate the molecular heterogeneity characteristic of breast cancer. As tumors become more heterogeneous, distinct subpopulations of cells within the tumor can evolve, leading to specialized gene modules that could contribute to therapeutic resistance. This modular organization could allow certain parts of the tumor network to avoid treatment intervention, allowing cancer to persist even under treatment [52,53,54,55]. The closeness centrality, which reflects the ease of communication between genes in a network, is notably higher in healthy networks. This suggests a more efficient and balanced distribution of regulatory interactions in non-cancerous tissue. In contrast, the lower value of this metric in breast cancer subtypes implies that gene communication within cancer networks is more localized and fragmented, potentially disrupting the coordination of crucial signaling pathways. This reduced value in cancerous tissues may contribute to the dysregulated regulatory dynamics observed in tumors, where key pathways such as DNA damage response, metabolic control, and immune evasion may no longer be effectively regulated, leading to chaotic and uncontrolled tumor growth [56,57].

Similarly, the reduction in degree (the number of connections per node) and global efficiency in cancer subtypes further emphasizes the loss of network connectivity and overall regulatory complexity. In healthy tissue, a higher degree suggests more robust gene connectivity, which is essential for maintaining genomic stability and efficient signal transduction across the entire network. However, as cancer progresses, the reduced number of connections in the network likely reflects a dysfunctional regulatory environment, where fewer key regulatory hubs remain active. This loss of connectivity could result in cancer cells relying on a smaller number of critical genes or hubs for survival, increasing the vulnerability of these nodes but also making the network more dependent on specific oncogenic drivers [27,58]. Lastly, the lower network density observed in all breast cancer subtypes, especially in Basal samples, which are the most aggressive breast cancer subtype, indicates that these networks are more fragmented. This fragmentation likely exacerbates the genomic instability characteristic of aggressive cancers, contributing to the rapid and uncontrolled growth of these tumors. The reduced cohesiveness of gene interactions in Basal subtype networks may reflect an underlying loss of coordinated regulation, allowing for increased mutational burden and more frequent chromosomal rearrangements, which drive the aggressive nature of this cancer subtype [55]. Together, these network metrics provide a broader view of how breast cancer subtypes reorganize their genomic interactions, probably to sustain malignant behavior, highlighting the structural adaptations that underpin their growth and survival.

The comparison between aggregated and single-sample co-expression networks, as illustrated by the Jaccard index in Figure 4, provides important insights into the topological coherence of these network types across normal and breast cancer subtypes. The lower similarity observed in normal tissue between its aggregated and individual networks suggests a higher degree of variability in gene co-expression patterns. This variability could be indicative of a broader range of regulatory mechanisms that maintain homeostasis in healthy tissue, as seen in studies where normal networks often exhibit complex dynamic regulatory landscapes [6,59]. In contrast, the higher Jaccard indices in cancer subtypes reflect a greater alignment between aggregated and single-sample networks, suggesting that these subtypes share more common co-expression modules. This increased coherence in cancer may point towards more defined oncogenic programs and conserved regulatory disruptions that support tumor growth and survival. However, the observed decline in similarity among individual networks underscores the inherent heterogeneity within tumor samples, reflecting variations in gene expression and regulatory dynamics that may arise due to genetic mutations, epigenetic alterations, or microenvironmental influences [60,61]. This heterogeneity aligns with the recognized genetic diversity and adaptive capabilities of cancer cells, which are known to exploit flexible co-expression patterns to evade therapeutic interventions. While aggregated networks capture the core regulatory frameworks, single-sample networks reveal unique co-expression patterns that contribute to the personalized complexity of each tumor, highlighting patient-specific genomic interactions that could inform individualized therapeutic strategies [30].

The results of our survival analysis indicate that the proportion of CIS interactions in breast cancer patients is not significantly related to the overall survival or 5-year survival in molecular subtypes. These results question the idea that intrachromosomal interactions, as represented by the proportion of CIS interactions can play a decisive role in determining the outcome of the patient (Figure 5). Our data do not reveal significant differences in survival between high and low CIS groups in all subtypes. This lack of connection may indicate that genomic stability, particularly in terms of intra-chromosome connectivity, is not a factor in the prediction of breast cancer patients. The slight trend observed in the Basal subtype, although not statistically significant, suggests a weak association that requires further investigation. Here, the concept of chromosomal instability (CIN) and its role in cancer progression provides a possible context for our findings. CIN, characterized by an increased rate of chromosomal gains and losses, is a hallmark of many cancers and contributes to tumor heterogeneity and evolution. It has been observed that while intermediate levels of CIN can promote tumorigenesis by increasing tumor heterogeneity, very high levels of CIN may impair tumor growth due to the generation of nonviable karyotypes [62,63]. Our results may be consistent with these observations, as high levels of CIS interactions, reflecting potential genomic stability, are not associated with better survival outcomes. This suggests that, rather than direct the proportion of CIS, the broader context of genomic alterations and their interactions, including the balance between genomic stability and instability, may play a more nuanced role in breast cancer progression and patient survival. The presence of stable CIS interactions, while reflecting a less chaotic chromosomal environment, may fail to capture the broader genomic instability that drives cancer progression through other mechanisms, such as translocations or structural rearrangements in key oncogenic regions. Additionally, while aneuploidy and CIN are common in many cancers, including breast cancer, their precise impact on survival remains complex and context-dependent [64,65], requiring a more integrative analysis of genomic and clinical data to fully understand their contributions to patient prognosis.

The analysis of high-degree genes and their cytoband localization in single-sample networks across breast cancer subtypes provides significant insights into the molecular heterogeneity and regulatory disruptions inherent to each subtype. The high recurrence of genes like C1QBP and RPS3A in Luminal A, USP31 and RRM1 in Luminal B, and PSMD8 and YME1L1 in Basal subtypes highlights the subtype-specific regulatory networks that underlie tumor progression [66,67,68,69,70]. In Luminal B, USP31 is involved in immune regulation, particularly in the modulation of the NF-κB pathway, which is frequently dysregulated in cancer. By stabilizing key components of this pathway, USP31 may enhance immune evasion and contribute to the tumor’s ability to escape immune surveillance [71]. RRM1 plays a crucial role in DNA synthesis and repair, making it essential for the high proliferative activity of cancer cells. Overexpression of RRM1 has been linked to chemoresistance in breast cancer, as it supports DNA replication in rapidly dividing tumor cells and promotes survival under genotoxic stress [68]. Also, in the Basal subtype, PSMD8 is involved in protein degradation via the ubiquitin–proteasome system, which is crucial for maintaining protein homeostasis in cancer cells. Dysregulation of the proteasome system can lead to the accumulation of misfolded proteins, promoting tumor progression and metastasis [69]. YME1L1, a key mitochondrial protease, plays a critical role in regulating mitochondrial dynamics, including the processes of fusion and fission, which are essential for maintaining cellular energy and metabolic homeostasis. Overexpression of YME1L1 in Basal breast cancer could contribute to the aggressive phenotype observed in this subtype by enhancing mitochondrial biogenesis and respiratory function. Given that disruptions in YME1L1 have been linked to impaired mitochondrial structure and increased production of reactive oxygen species (ROS), its upregulation in Basal tumors may support the high metabolic demands and proliferative capacity of these cancer cells [72].

These high-degree genes are pivotal in maintaining the integrity of the co-expression network, as their centrality in the largest network components indicates their crucial role in regulating multiple pathways. C1QBP has been shown to play a role in cell metabolism and immune evasion, which are critical factors in cancer cell survival and proliferation. Elevated C1QBP expression has been linked to poor prognosis in several cancers, due to its role in promoting cell proliferation and metastasis by modulating mitochondrial function [66]. RPS3A is a component of the ribosomal machinery, and its overexpression is associated with increased protein synthesis, a hallmark of rapidly dividing cancer cells. RPS has been implicated in oncogenic transformation and may contribute to the aggressive growth observed in Luminal A breast cancer by promoting cell cycle progression [67].

The cytoband localization of these high-degree genes further underscores their significance in breast cancer biology. Regions such as 11q13.1 and 17q12.1 in Luminal subtypes are known to harbor oncogenes like CCND1, which drives cell cycle progression and is frequently amplified in breast cancer [33,34,35,73]. The amplification of 11q13.1 has been linked to increased tumor proliferation and worse prognosis in patients. Similarly, the frequent localization of high-degree genes in 8q24.3, a region known for MYC amplification, suggests that this hotspot plays a pivotal role in the aggressive behavior of Luminal B and Her2 subtypes [74,75]. MYC is a master regulator of cell growth and metabolism, and its dysregulation is a hallmark of many cancers, including breast cancer. Its amplification in these subtypes could drive rapid tumor growth and contribute to poor clinical outcomes. For the Basal subtype, frequent cytoband localization in 6p21.1 and 19q13.2 correlates with regions involved in immune modulation and cellular adhesion, crucial processes in the metastatic potential of this aggressive subtype [76,77]. These findings emphasize the importance of cytobands such as 11q13.1, 8q24.3, and 17q12.1 as recurrent regions across multiple subtypes, reflecting shared vulnerabilities in the genomic landscape of breast cancer. Yet, the distinct genetic contexts of each subtype underscore the complexity of their molecular heterogeneity, as evidenced by the differential distribution of high-degree genes. This variation in genomic disruptions and the localization of high-degree genes within key cytobands highlights their functional roles in driving subtype-specific oncogenic processes and potentially influencing treatment resistance.

The identification of patient-specific co-expression networks through single-sample analyses represents a pivotal strategy for advancing precision medicine. By capturing the unique genomic profiles of each patient, this approach reveals the specific regulatory disruptions driving tumor progression in different breast cancer subtypes. Unlike aggregated analyses, which may obscure individual variability, single-sample networks highlight the distinct vulnerabilities of each tumor, enabling a more tailored understanding of the molecular mechanisms at play. In precision oncology, this methodology allows for the identification of key network nodes that could serve as actionable therapeutic targets. By focusing on the unique structural disruptions within an individual’s network, clinicians can prioritize interventions that target these critical hubs, increasing the likelihood of therapeutic efficacy. For example, network hubs with higher connectivity or genomic instability could be targeted with drugs that specifically modulate those pathways, offering a more precise intervention strategy [14].

Moreover, these individualized networks provide insights into potential biomarkers for predicting treatment response or resistance. Patients whose networks exhibit specific disruptions—such as increased intrachromosomal interactions or fragmentation—could be stratified for targeted therapies designed to address those specific genomic irregularities. By leveraging this personalized approach, precision medicine can optimize treatment decisions, reducing unnecessary treatments and improving outcomes by tailoring therapies to the unique molecular landscape of each patient’s tumor. Thus, integrating single-sample network analysis into clinical practice offers a promising path toward more effective and personalized cancer treatment strategies [14].

## 4. Materials and Methods

### 4.1. Data Acquisition and Pre-Processing

Gene expression data (RNA-Seq) from healthy and breast cancer samples were obtained using the TCGAbiolinks (v2.25.3) package in R environment (v4.2.1) from The Cancer Genome Atlas (TCGA) initiative [78]. We retrieved 1231 transcriptome profiling datasets categorized under “Gene Expression Quantification”, processed using the STAR workflow. Additional gene annotations, including chromosome names, gene types, GC content, and start/end positions, were retrieved from BioMart (v2.54.1) [79].

Expression matrices for each molecular subtype of breast cancer, Luminal A (*N* = 565), Luminal B (*N* = 217), Her2 (*N* = 82), Basal (*N* = 192), and control (*N* = 113) breast tissue, were prepared for analysis according to the following steps. Initially, low-expressed genes were filtered by excluding those within the lower 0.25 quantile, those with zero expression across samples, and those with an average normalized count (TMM) below 50. We then accounted for potential batch effects and technical variability due to multi-site sample processing and library preparation. To mitigate these biases, normalization was performed using the EDASeq (v2.32.0) [80] and NOISeq (2.42.0) [81] packages. This process involved within-lane normalization for transcript length and GC content, followed by between-lane normalization using the Trimmed Mean of M-values (TMM) method to regularize read depth across samples. Furthermore, to address batch effects and potential noise introduced during sequencing or sample handling, we applied the ARSyNseq [82] noise reduction algorithm. This approach, integrated with the NOISeq package, specifically removes systematic variability and enhances the separation of molecular subtypes. The effectiveness of these steps was evaluated and confirmed visually using Principal Component Analysis (PCA), indicating successful mitigation of technical biases and batch effects.

### 4.2. Gene Co-Expression Network Inference

We implemented the ARACNe algorithm [83] to infer GCNs for our data, which calculates the mutual information (MI) between gene expression profiles to identify significant coexpression connections. Specifically, ARACNe detects nonlinear relationships between gene pairs by estimating their mutual information, ensuring that only the strongest direct gene–gene interactions are retained. A multi-core version of ARACNe [49] was used to enhance computational efficiency, leveraging parallel processing across multiple cores. For each subtype and normal tissue, the GCN was computed for approximately 12.5 million possible interactions between 5000 genes (https://github.com/josemaz/aracne-multicore (accessed on 10 November 2024)). To ensure comparability across networks, we selected the top 10,000 gene interactions for further analysis. This threshold was selected based on its capacity to maintain the same network size across all conditions, enabling a robust comparison of network characteristics. Previous research has demonstrated that analyzing this number of interactions allows for capturing both structural and functional network properties effectively [17,53,54,84]. Sensitivity analyses in these references confirmed the stability of findings at this threshold, supporting its use in our study.

### 4.3. Inference of Single-Sample Co-Expression Networks

We inferred individual co-expression networks for each breast cancer sample using the LIONESS equation [12]. This approach iteratively excludes one sample at a time from the full cohort, constructing GCNs both with and without the excluded sample. The difference between these two networks reveals the unique co-expression structure for each sample. Then, we calculated MI, to quantify the statistical dependencies between gene pairs, forming the foundational connections of these networks. To enhance computational efficiency, we integrated LIONESS with a parallelized ARACNe implementation, allowing MI to be calculated across multiple cores, thus inferring each individual network in a sequential manner. This implementation can be found at https://github.com/PatricioLOPSA/LIONESS-MI.git (accessed on 10 November 2024)). Each single-sample network was subsequently analyzed by focusing on the top 10,000 edges, ranked by their LIONESS-MI scores, which highlighted the most significant co-expression interactions.

### 4.4. Intra-Chromosomal Proportion of Co-Expression Networks

Previous research has shown that GCNs in breast cancer tend to have more interactions among genes located on the same chromosome (CIS), while networks from non-cancerous breast tissue exhibit a higher prevalence of interactions between genes on different chromosomes (TRANS) [6,27]. To explore whether these patterns are consistent across different molecular subtypes of breast cancer, we analyzed both aggregated networks and single sample networks to determine CIS- and TRANS-interaction proportions in each breast cancer subtype and in normal breast tissue. For this analysis, we focused on the top 10,000 gene–gene interactions in each network to ensure consistency in network size and to allow robust comparison of CIS-/TRANS- interaction proportions across both aggregated and single-sample networks. Additionally, we used Cytoscape v3.10 to visualize network topology and interaction patterns (especially in aggregated networks) [85], providing a comprehensive view of the differences between cancerous and normal breast tissue.

We conducted a comprehensive network metric analysis to investigate the structural properties of GCNs for both aggregated and single-sample networks across all breast cancer molecular subtypes and normal breast tissue. Our focus was on the largest connected component of each network to ensure consistent and robust metric evaluations. Key metrics, such as clustering coefficient, modularity, global efficiency, density, closeness, centrality, and degree, were calculated to capture topological organization, functional modularity, and connectivity. Aggregated networks were analyzed using the top 10,000 gene–gene interactions ranked by MI values, while for SSNs, LIONESS-MI was employed with absolute values to derive comparable interactions. This dual approach allowed us to systematically evaluate network topology across different breast cancer subtypes and normal tissue, revealing distinct characteristics in each network type.

### 4.5. Survival Analysis

The survival analysis was conducted using Kaplan–Meier estimations to compare overall survival between median-high and median-low CIS (intrachromosomal interaction) groups across different breast cancer subtypes. Clinical data were merged with CIS group information, and survival outcomes were assessed for both overall survival and a 5-year survival endpoint (1825 days). Survival curves were generated using the Kaplan–Meier method, with statistical significance evaluated by the log-rank test. Additionally, Cox proportional hazard models were employed to estimate hazard ratios between high and low CIS groups. Visualizations included confidence intervals and risk tables for each group, enhancing the interpretability of differences in survival probability. The results were plotted using the “ggsurvplot” function from the survminer R package version 4.4.2.

## 5. Conclusions

In this study, we performed an in-depth analysis of gene co-expression networks in breast cancer, utilizing both aggregated and single-sample approaches to uncover structural and functional genomic alterations across different molecular subtypes. Our findings reveal that breast cancer subtypes exhibit significant fragmentation of co-expression networks, characterized by a change from interchromosomal to intrachromosomal interactions. This transition may suggest a disruption of long-range genomic communication, leading to localized genomic regulation and increased genomic instability. The analysis of single-sample networks reinforces these observations, demonstrating that the reconfiguration toward CIS interactions and specific cytoband disruptions is consistent at the individual patient level. This highlights the molecular heterogeneity inherent in breast cancer and underscores the importance of personalized network analyses in capturing tumor-specific genomic architectures.

Despite the pronounced alterations in network topology and interaction patterns, our survival analyses indicate that the proportion of CIS interactions does not significantly correlate with patient outcomes across the breast cancer subtypes studied. This suggests that while genomic instability is a hallmark of cancer, the prognostic value of CIS interaction proportions alone may be limited. Furthermore, the identification of high-degree genes and their localization within critical cytobands provides insights into subtype-specific regulatory networks and potential therapeutic targets. These genes, central to the co-expression networks, may play pivotal roles in oncogenic processes and represent avenues for targeted interventions.

Overall, our study advances the understanding of the genomic architecture in breast cancer by highlighting the significance of network fragmentation, localized genomic regulation, and molecular heterogeneity. The application of single-sample co-expression network analysis proves to be a powerful tool in uncovering individual-specific genomic interactions, opening the way for personalized therapeutic strategies. Future research integrating comprehensive genomic alterations and exploring their functional implications will be crucial in developing effective interventions and improving patient outcomes in breast cancer.

## Figures and Tables

**Figure 1 ijms-25-12163-f001:**
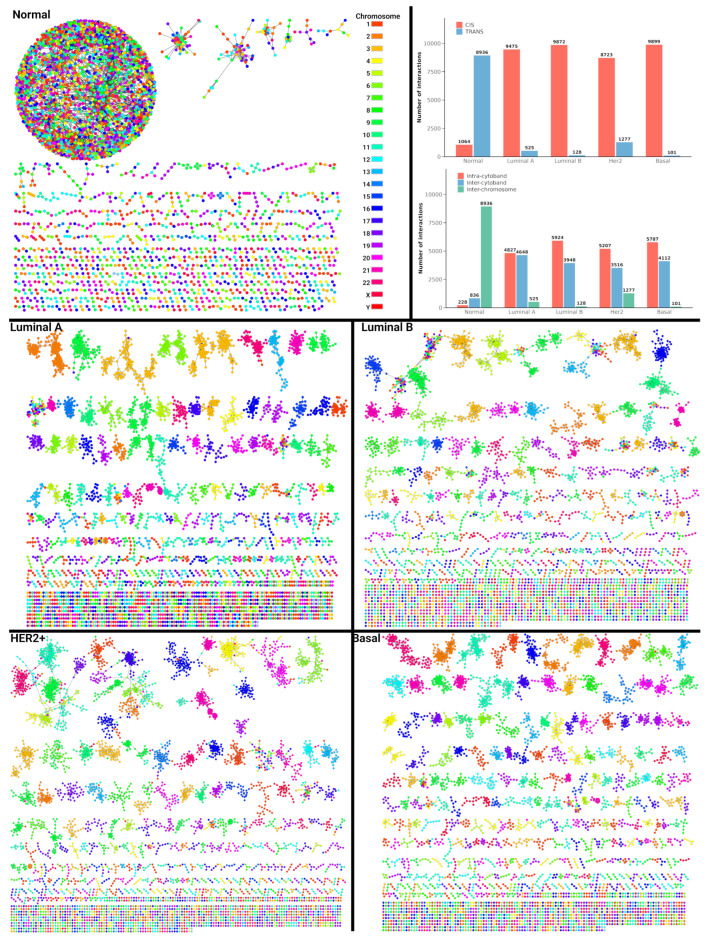
Visualization of gene co-expression in healthy breast tissue and breast cancer subtypes (Top—10,000 higher edges). Genes are color-coded by chromosome, highlighting the distribution of interchromosomal and intrachromosomal interactions. Upper right, evaluation of CIS/TRANS and Inter/Intracytoband interactions in healthy breast tissue and breast cancer subtypes.

**Figure 2 ijms-25-12163-f002:**
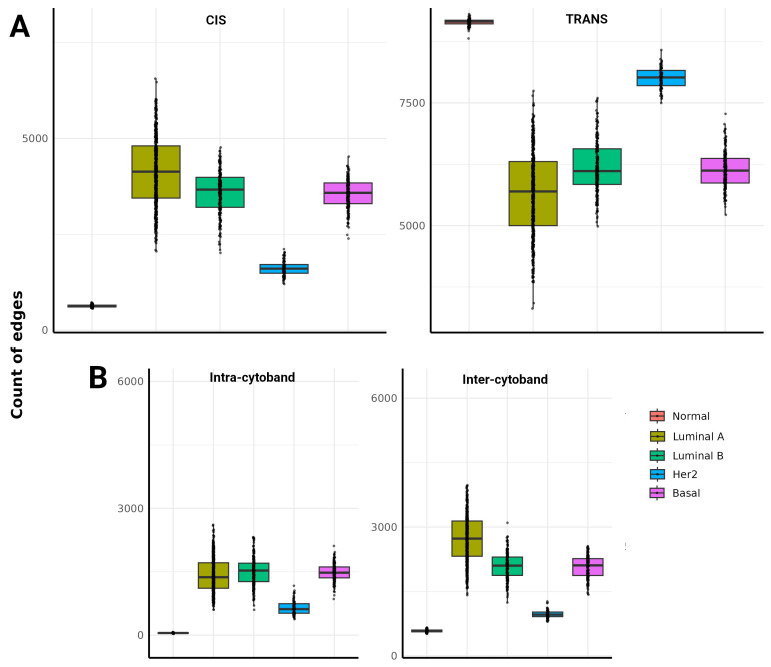
Boxplots of CIS/TRANS (**A**) and Inter/Intra-cytoband (**B**) interactions in healthy breast tissue and breast cancer subtypes across all samples (single-samples). Each point corresponds to the number of edges in a single-sample network. Notice that the distribution of control samples is much narrower than cancer subtypes.

**Figure 3 ijms-25-12163-f003:**
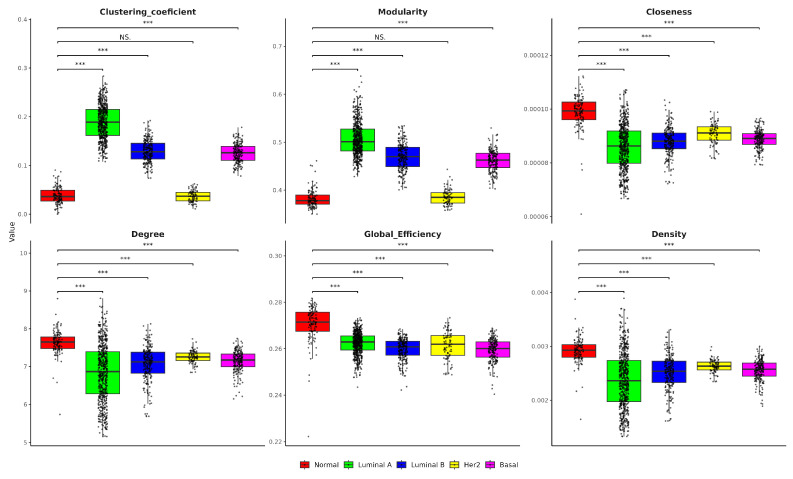
Network metrics in the largest component (clustering coefficient, modularity, closeness, degree, global efficiency, density) for healthy (normal) and breast cancer subtypes across all samples. *** *p*-value < 0.001. NS = Non-Significant.

**Figure 4 ijms-25-12163-f004:**
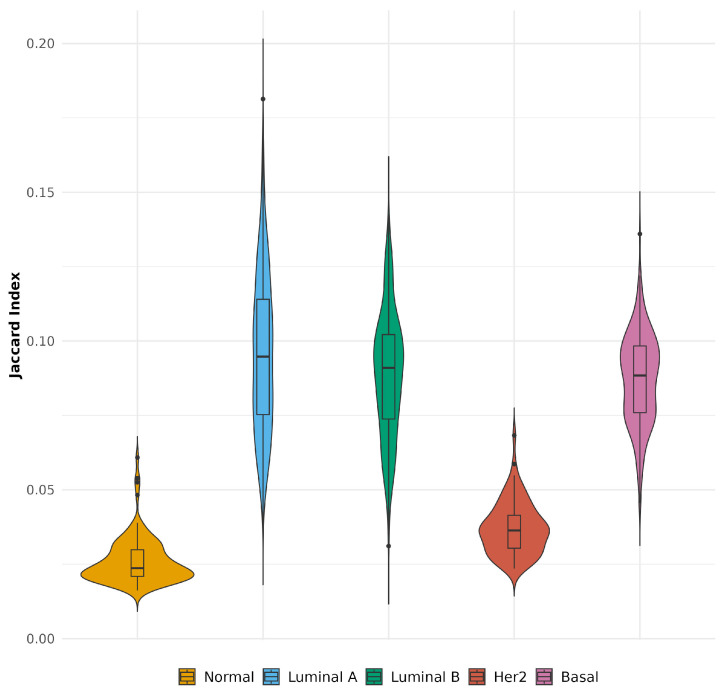
Comparison between the top 10,000 interactions from aggregated networks with the top 10,000 interactions from each single sample network using the Jaccard index.

**Figure 5 ijms-25-12163-f005:**
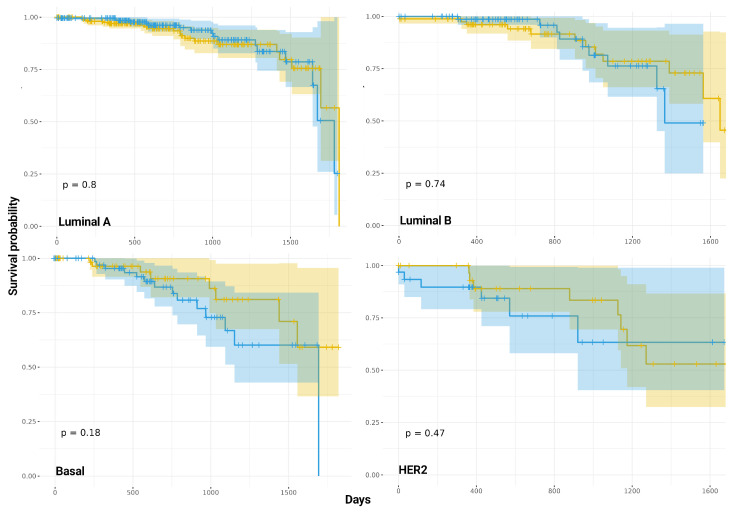
Kaplan–Meier survival curves for breast cancer patients with high (blue) and low (yellow) of CIS interactions across different molecular subtypes (Luminal A, Luminal B, Her2 and Basal).

**Figure 6 ijms-25-12163-f006:**
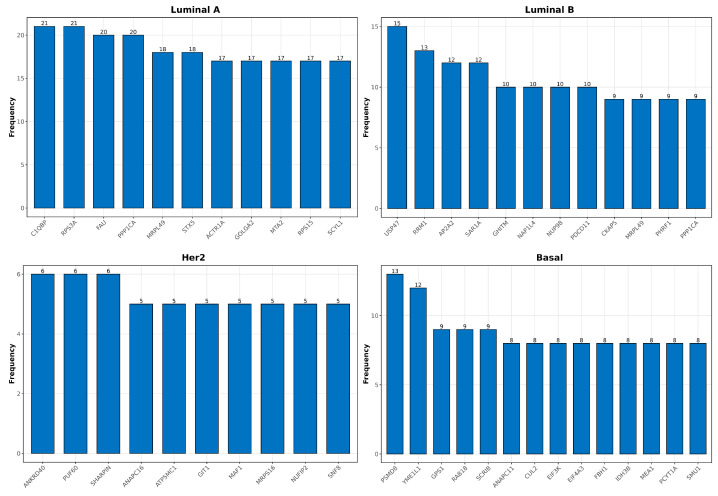
Distribution of High-Degree genes in Single-Sample co-expression networks across breast cancer subtypes.

**Figure 7 ijms-25-12163-f007:**
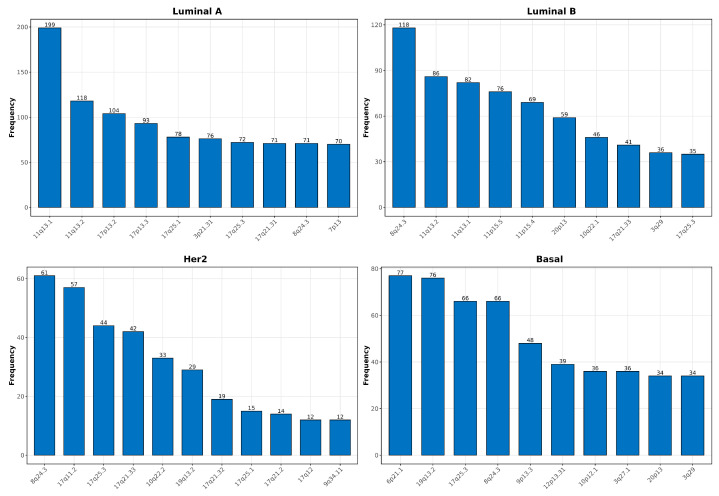
Top 10 High-Degree genes in breast cancer subtypes: Cytoband localization across Single-Sample networks.

**Table 1 ijms-25-12163-t001:** Chromosomal localization of major intrachromosomal and intracytoband interactions.

Subtype	Chromosome	Frequency	Proportion
	1	223	0.000105
Healthy	19	98	0.000090
	2	89	0.000114
	17	1742	0.002479
Luminal A	11	1601	0.001847
	8	871	0.003520
	11	1612	0.001860
Luminal B	17	1530	0.002177
	8	792	0.003201
	17	2047	0.002913
Her2	8	754	0.003047
	11	698	0.000805
	1	1013	0.000475
Basal	19	749	0.000687
	10	641	0.002396

**Table 2 ijms-25-12163-t002:** Top-three intra-cytoband interactions for the five phenotypes under study.

Subtype	Cytoband	Frecuency	Proportion
	6p21.32	17	0.010650
Healthy	8p11.23	13	0.068420
	8q24.3	7	0.001210
	11q13.1	264	0.051250
Luminal A	8q24.3	206	0.035650
	11q13.2	151	0.062530
	8q24.3	377	0.065247
Luminal B	17q11.2	227	0.066706
	11q13.1	205	0.039798
	17q11.2	367	0.107850
Her2	8q24.3	331	0.057290
	17q25.3	325	0.061870
	6p21.1	272	0.066420
Basal	8q24.3	224	0.038770
	17q25.3	182	0.034650

## Data Availability

The code to perform all calculations presented here can be found at https://github.com/rponce20/BrCA_SSN_TCGA (accessed on 10 November 2024).

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
