# Peer review of "Single-Sample Networks Reveal Intra-Cytoband Co-Expression Hotspots in Breast Cancer Subtypes"

_ijms, 2024, doi:10.3390/ijms252212163_

Round 1
Reviewer 1 Report
Comments and Suggestions for Authors
The manuscript presents an interesting study exploring the use of single-sample gene co-expression network analysis to investigate genomic alterations across breast cancer subtypes. The application of the LIONESS algorithm to construct individual networks provides valuable insights into the molecular heterogeneity of breast cancer and highlights potential subtype-specific regulatory networks.
However, there are several areas where the manuscript could be improved to enhance its clarity, reproducibility, and impact:
1. While the authors mention the use of ARACNe and LIONESS algorithms, the specific parameters and settings used during network inference are not fully detailed. Providing information on parameters such as mutual information thresholds, bootstrapping iterations, and any modifications made to default settings would improve reproducibility.
2. The choice of focusing on the top 10,000 gene interactions is mentioned but not adequately justified. It would be beneficial to explain how this threshold was determined and whether sensitivity analyses were conducted to assess the impact of this choice on the results.
3. Elaborate on the preprocessing steps, including the criteria for filtering low-expression genes, normalization methods, and how batch effects or technical variabilities were addressed. Including references to the specific functions or packages used would be helpful.
4. Some figures, particularly network visualizations (e.g., Figures 1 and 2), may be complex and difficult to interpret. Enhancing the clarity of these figures by increasing resolution, simplifying visuals, or highlighting key elements will aid reader comprehension. Ensure all figures have descriptive captions that clearly explain what is being shown.
5. In several sections, results are briefly mentioned without sufficient explanation or context. For example, when discussing network metrics or high-degree genes, provide more detailed interpretations of what these results imply for breast cancer biology.
6. While the identification of high-degree genes and critical cytobands is valuable, the manuscript would benefit from a deeper biological interpretation. Discuss the known functions of these genes and how they may contribute to breast cancer development or progression. Linking your findings to existing literature can provide context and underscore their significance.
7. Highlight how the identified subtype-specific genes and network disruptions could inform potential therapeutic strategies or biomarkers for personalized medicine.
8. While the English language quality is acceptable, there are minor grammatical errors and awkward phrasings that could be improved. Careful proofreading or assistance from a professional editor would enhance readability.
Author Response
Dear Reviewers,
We would like to extend our sincere thanks for your thoughtful and constructive feedback on our manuscript. We have carefully considered each of your suggestions and have made the necessary revisions to improve the quality of our work in accordance with those recommendations. A revised version of the manuscript, as well as a tracked changes version are attached for your review.
We are confident that these revisions address all the insights provided by both reviewers and significantly enhance the manuscript’s overall quality. We believe the work now offers a substantial contribution to the field of breast cancer genomics. Below, we provide a point-by-point response to your comments. For clarity, our responses are highlighted in bold.
Thank you once again for your time and consideration.
Jesús Espinal-Enríquez
On behalf of all authors.
__________________________________
Reviewer 1
Comments and Suggestions for Authors.
The manuscript presents an interesting study exploring the use of single-sample gene co-expression network analysis to investigate genomic alterations across breast cancer subtypes. The application of the LIONESS algorithm to construct individual networks provides valuable insights into the molecular heterogeneity of breast cancer and highlights potential subtype-specific regulatory networks.
However, there are several areas where the manuscript could be improved to enhance its clarity, reproducibility, and impact:
- While the authors mention the use of ARACNe and LIONESS algorithms, the specific parameters and settings used during network inference are not fully detailed. Providing information on parameters such as mutual information thresholds, bootstrapping iterations, and any modifications made to default settings would improve reproducibility.
Following your recommendation, we have revised the methodology section to specify the parameters used during network inference. To clarify, as we analyzed all possible gene pairs from our expression matrices (each containing over 5,000 genes), we did not perform mutual information thresholding. Instead, we retained the top 10,000 interactions, as each individual network has a distinct distribution of mutual information values. This selection allowed us to standardize network size, facilitating meaningful comparisons across networks. Additionally, we used the other ARACNe’s default settings for the analysis.
- The choice of focusing on the top 10,000 gene interactions is mentioned but not adequately justified. It would be beneficial to explain how this threshold was determined and whether sensitivity analyses were conducted to assess the impact of this choice on the results.
Thank you for highlighting this point. We based our decision to focus on the top 10,000 interactions on extensive prior studies, which demonstrate that networks within the 10,000-100,000 strongest interactions reliably capture the relevant biological features. Larger networks tend to incorporate less biologically meaningful connections, while smaller ones risk losing essential information. Additionally, we have several publications where we perform network inference with varying sizes, showing that biological phenomena, such as the loss of interchromosomal co-expression, as well as enriched biological processes, remain statistically significant across different network sizes. Thus, we selected 10,000 interactions to balance information retention with biological relevance, enhancing comparability across networks.
In the revised version of this manuscript, the methodology to construct the co-expression networks is as follows:
“In our study, we focused on the top 10,000 gene interactions to ensure comparability across all gene co-expression networks (GCNs). This threshold was selected based on its capacity to maintain the same network size across all conditions, enabling a robust comparison of network characteristics. Previous research has demonstrated that analyzing this number of interactions allows for capturing both structural and functional network properties effectively (Alcalá-Corona et al., 2016, 2017; Velazquez-Caldelas et al., 2019; Zamora-Fuentes et al., 2020). Sensitivity analyses in these references confirmed the stability of findings at this threshold, supporting its use in our study.”
- Elaborate on the preprocessing steps, including the criteria for filtering low-expression genes, normalization methods, and how batch effects or technical variabilities were addressed. Including references to the specific functions or packages used would be helpful.
We have improved and extended the description of the processing data method. The revised version of our manuscript contains these paragraphs:
“Gene expression data (RNA-Seq) from healthy and breast cancer samples were obtained using the TCGAbiolinks (v2.25.3) package in R environment (v4.2.1) from The Cancer Genome Atlas (TCGA) initiative[17]. We retrieved 1,231 transcriptome profiling datasets categorized under "Gene Expression Quantification," processed using the STAR workflow. Additional gene annotations, including chromosome names, gene types, GC content, and start/end positions, were retrieved from BioMart (v2.54.1)[18].
Expression matrices for each molecular subtype of breast cancer, Luminal A (N=565), Luminal B (N=217), Her2 (N=82), Basal (N=192), and control (N=113) breast tissue, were prepared for analysis according to the following steps. Initially, low-expressed genes were filtered by excluding those within the lower 0.25 quantile, those with zero expression across samples, and those with an average normalized count (TMM) below 50. We then accounted for potential batch effects and technical variability due to multi-site sample processing and library preparation. To mitigate these biases, normalization was performed using the EDASeq (v2.32.0) [19] and NOISeq (2.42.0)[20] packages. This process involved within-lane normalization for transcript length and GC content, followed by between-lane normalization using the Trimmed Mean of M-values (TMM) method to regularize read depth across samples. Furthermore, to address batch effects and potential noise introduced during sequencing or sample handling, we applied the ARSyNseq [21] noise reduction algorithm. This approach, integrated with the NOISeq package, specifically removes systematic variability and enhances the separation of molecular subtypes. The effectiveness of these steps was evaluated and confirmed visually using Principal Component Analysis (PCA), indicating successful mitigation of technical biases and batch effects.”
- Some figures, particularly network visualizations (e.g., Figures 1 and 2), may be complex and difficult to interpret. Enhancing the clarity of these figures by increasing resolution, simplifying visuals, or highlighting key elements will aid reader comprehension. Ensure all figures have descriptive captions that clearly explain what is being shown.
We have improved the resolution of all figures. We hope that the merged pdf retains the high resolution with which the manuscript was compiled. Additionally, we also modified some figure captions in order to make the description clearer.
- In several sections, results are briefly mentioned without sufficient explanation or context. For example, when discussing network metrics or high-degree genes, provide more detailed interpretations of what these results imply for breast cancer biology.
We have incorporated a more detailed discussion in several points of our results. For clarity, in what follows we will describe the paragraphs and lines in which we modified the discussion.
Regarding the differences in network topology for individuailzed co-expression networks and loss of trans interactions in cancer:
Lines 337 - 360
The findings from our co-expression network analysis highlight significant differences in network topology between healthy breast tissue and breast cancer subtypes, reflecting underlying molecular heterogeneity and genomic instability. In healthy tissue, the presence of a large giant component with extensive interchromosomal interactions suggests a highly coordinated genomic architecture essential for maintaining normal cellular function. In contrast, the fragmentation and prevalence of intrachromosomal interactions observed in cancer networks suggest disrupted long-range genomic communication. This pattern aligns with previous studies that have documented a loss of TRANS-co-expression in cancer tissues, which is thought to contribute to the breakdown of normal regulatory mechanisms and promote localized genomic instability[26–29].
The change from the predominant TRANS interaction to the CIS and the increase in intra- and intercytoband interactions observed in cancer subtypes (Figure 2) accentuate the idea of disruption of long-term genomic communication and the transition to a more local regulatory framework, which reflects the fundamental decomposition of genomic architecture coordination, and is consistent with recent studies where such fragmentation and increased intrachromosome interactions are associated with genomic instability and local chromosome rearrangements in cancer[30–32]. These localized interactions, especially the increase in intra- and inter cytoband interactions, indicate a transition towards chromosome instability where certain genomic regions become hotspots for mutations and rearrangements, facilitating oncogenic transformation[33,34]. This interruption of long-term interactions can lead to the formation of transcriptionally active domains, which may promote specific cancer gene expression patterns and contribute to the aggressive behavior and heterogeneity observed in breast cancer subtypes[35]. Moreover, the accumulation of these localized interactions could create genomic regions with sustained aberrant expression, enhancing cellular plasticity and adaptability in response to external stresses or therapeutic interventions. This emphasizes the dynamic reorganization of chromosomal regions as a hallmark of cancer progression. The results highlight the importance of understanding the genomic spatial organization in cancer.
Regarding the implications of loss of trans interactions and the concomitant gain of intra-chromosomal links:
Lines 361 - 370
Similarly, the results of the single-sample co-expression network analysis of the balance of CIS and TRANS interactions between healthy and cancerous tissues showed that while the initial aggregated network presented a marked increase in CIS interactions within breast cancer subtypes, the strength of the current results lies in the individualized analysis of single-sample co-expression networks. Remarkably, even at the single-sample level, the CIS/TRANS interaction patterns remain consistent with the aggregated network findings, supporting the robustness of these trends across different levels of analysis [24,27,36,37]. This provides compelling evidence that the disruption of long-range genomic communication is not an artifact of network aggregation, but may be considered a fundamental feature of the genomic instability of cancer. This transition toward increased CIS interactions likely reflects localized genomic alterations that contribute to chromosomal rearrangements or focal amplifications, which are common features of cancer. This rearrangement may foster genomic regions that become transcriptionally active or prone to instability, thus favoring tumor heterogeneity and progression by enabling localized oncogenic signaling or mutational hotspots.
Regarding the improvement in resolution of individualized gene co-expression networks
Línea 371 - 388
Importantly, the individualized nature of this analysis resolves many of the limitations associated with aggregated network studies, where data from multiple samples are combined, potentially masking the unique genomic interactions that occur within individual tumors. In this regard, single-sample networks offer a more precise view of tumor-specific genomic coordination and the localized genomic disruptions characteristic of different types of cancer [12,15,16]. This individual-based approach is crucial, as genomic heterogeneity is a well-established feature of cancer, and aggregated analyses often fail to capture the full extent of variability between tumors [38]. Studies of individual genomic profiles have demonstrated that single-sample analyses are more effective at identifying clinically relevant mutations and genomic alterations that drive tumor behavior, providing a clearer path for personalized therapeutic interventions [39,40]. The increase in CIS interactions and the reconfiguration of co-expression networks toward intra- and inter cytoband interactions supports the notion of increased genomic instability. These changes align with prior research showing that cancer cells often reorganize their chromatin structures to enhance localized genomic interactions, promoting the formation of transcriptional hubs that regulate oncogene expression and contribute to tumor growth [28,30,37]. Furthermore, capturing heterogeneity in coexpression patterns at the individual level would enable the identification of unique genomic alterations that could foster the development of tailored therapeutic strategies. Thus, integrating genomic analyses of individual samples into oncology research may provide deeper insight into underlying genomic instability and help identify potential biomarkers or novel patient-specific therapeutic targets.
Regarding the calculations of hotspots of intra-chromosomal interactions in aggregated and individual networks
Lines 389 - 422
In this study, we conducted two types of co-expression network analyses: aggregated networks and single-sample networks. Aggregated networks were constructed by combining gene expression data from multiple samples within each breast cancer subtype and healthy tissue, producing a collective overview of genomic interactions in five networks—one for each breast cancer subtype (Luminal A, Luminal B, Her2, Basal) and one for the normal tissue phenotype. The goal was to identify the most frequent intra-chromosomal and intra-cytoband interactions across these combined networks. In contrast, single-sample networks were built for each patient, focusing on capturing unique genomic interactions at the individual level. Here, we analyzed which chromosomes and cytobands exhibited the highest number of intra-chromosomal and intra-cytoband interactions across multiple individual samples. This dual approach, reflected in the results summarized in Tables 1 and 2, allowed us to compare broad, population-level disruptions with patient-specific genomic rearrangements, highlighting similarities and distinctions between aggregated and single-sample networks.
Furthermore, Table 1 and Table 2 underscore the importance of understanding the regulatory hotspots that differentiate healthy breast tissue from breast cancer subtypes through the marked prevalence of intrachromosomal and cytoband interactions with both approaches. The intrachromosomal interactions, as shown in these tables, reveal that chromosome 1 plays a critical role in maintaining genomic stability in healthy samples, likely due to its size and its involvement in essential cellular processes, such as metabolism and DNA repair[41]. In contrast, cancerous tissues exhibit distinct chromosomal preferences; in Luminal subtypes, chromosomes 11, 17, and 1 showed greater interaction frequency, likely reflecting how cancer cells reorganize chromatin to enhance localized regulatory interactions[24,36,42–44]. This aligns with existing literature that identifies chromosomes 17 and 11 as regions frequently involved in breast cancer progression, primarily due to the presence of oncogenes like ERBB2 (Her2) and CCND1, which are known drivers of cell proliferation and tumor growth through gene amplification[45,46]. The prevalence of chromosome 17 in Her2 and Basal subtypes also correlates with the more aggressive clinical behavior of these cancers, as this region contains key genes involved in DNA repair and cell cycle control, whose dysregulation contributes to rapid tumor development[43,47].
In terms of cytoband interactions, healthy tissue samples demonstrate a strong predominance of interactions in cytoband 6p21.32, highlighting the role of immune-related genes, particularly those associated with the major histocompatibility complex (MHC), in preserving genomic integrity and preventing cancer development[48]. By contrast, the increased interactions within cytoband 11q13.1 in Luminal A and B subtypes are indicative of dysregulation in regions harboring CCND1, a well-known regulator of the cell cycle, which is often amplified in breast cancer and drives abnormal cell proliferation[49,50]. Cytoband 8q24.3, enriched in Luminal B and Her2 subtypes, is another hotspot for oncogene amplification, specifically for MYC, which plays a pivotal role in controlling cell growth and metabolism and is associated with poor prognosis in these subtypes due to its role in driving aggressive tumor behavior[51,52]. The continued presence of interactions in cytobands 17q11.2 and 8q24.3 in Her2 and Basal subtypes underscores the shared genomic vulnerabilities of these subtypes, as both regions are linked to critical oncogenes like ERBB2 and MYC, whose amplification exacerbates genomic instability, promoting resistance to therapy and aggressive tumor progression[53,54]. Additionally, Basal subtypes exhibit notable interactions in cytoband 6p21.1, suggesting that inflammatory and immune-related pathways contribute to the aggressive nature of this subtype, compounding the already complex genomic instability present in these tumors[55,56].
Regarding the network metrics of individualized networks and their implications:
Lines 423 - 450
Concerning the structural analysis of the single networks presented in Figure 3, which provides critical insights into the molecular organization of breast cancer subtypes, key differences in network metrics highlighted distinct genomic topology patterns that differentiate healthy tissues from cancer subtypes. The clustering coefficient, which reflects local connectivity in the network, is significantly higher in Luminal A, Luminal B, and Basal subtypes compared to healthy tissues. This increase suggests that cancer subtypes may form denser local interactions, potentially representing tightly coordinated gene modules that drive oncogenic signaling. In this context, higher clustering often indicates that genes within these networks are highly co-expressed, forming clusters of genes that may promote cancer progression through the formation of abnormal regulatory loops that can contribute to maintaining the proliferative advantage of cancer cells by reinforcing the robustness of pathways related to mechanisms such as cell cycle regulation, growth factor signaling, or resistance to apoptosis. Such dense clusters may reflect a change towards tumor-promoting interactions, enabling subtypes like Luminal B or Basal to evade the normal checkpoints that restrict cell growth[23,27,57,58].
Modularity, a measure of the degree of network division into different gene communities, is another key metric that distinguishes cancer subtypes. The significantly higher modularity observed in the Luminal A, Luminal B, and Basal subtypes suggests that these tumors develop specialized genetic interaction networks that could allow tumor cells to adapt to the stress of the tumor microenvironment. By forming well-defined substructures, these modules may facilitate the activation of oncogenic pathways essential to tumor survival and proliferation. Furthermore, this increase in modularity may also indicate the molecular heterogeneity characteristic of breast cancer. As tumors become more heterogeneous, distinct subpopulations of cells within the tumor can evolve, leading to specialized gene modules that could contribute to therapeutic resistance. This modular organization could allow certain parts of the tumor network to avoid treatment intervention, allowing cancer to persist even under treatment[59-63]. The closeness centrality, which reflects the ease of communication between genes in a network, is notably higher in healthy networks. This suggests a more efficient and balanced distribution of regulatory interactions in non-cancerous tissue. In contrast, the lower value of this metric in breast cancer subtypes implies that gene communication within cancer networks is more localized and fragmented, potentially disrupting the coordination of crucial signaling pathways. This reduced value in cancerous tissues may contribute to the dysregulated regulatory dynamics observed in tumors, where key pathways such as DNA damage response, metabolic control, and immune evasion may no longer be effectively regulated, leading to chaotic and uncontrolled tumor growth[64,65]. Similarly, the reduction in degree (the number of connections per node) and global efficiency in cancer subtypes further emphasizes the loss of network connectivity and overall regulatory complexity. In healthy tissue, a higher degree suggests more robust gene connectivity, which is essential for maintaining genomic stability and efficient signal transduction across the entire network. However, as cancer progresses, the reduced number of connections in the network likely reflects a dysfunctional regulatory environment, where fewer key regulatory hubs remain active. This loss of connectivity could result in cancer cells relying on a smaller number of critical genes or hubs for survival, increasing the vulnerability of these nodes but also making the network more dependent on specific oncogenic drivers[24,66]. Finally, the lower network density observed in all breast cancer subtypes, especially in Basal samples, which are the most aggressive breast cancer subtype, indicates that these networks are more fragmented. This fragmentation likely exacerbates the genomic instability characteristic of aggressive cancers, contributing to the rapid and uncontrolled growth of these tumors. The reduced cohesiveness of gene interactions in Basal subtype networks may reflect an underlying loss of coordinated regulation, allowing for increased mutational burden and more frequent chromosomal rearrangements, which drive the aggressive nature of this cancer subtype[63]. Together, these network metrics provide a comprehensive view of how breast cancer subtypes reorganize their genomic interactions, probably to sustain malignant behavior, highlighting the structural adaptations that underpin their growth and survival.
Regarding the comparison of structural properties between aggregated and individual networks:
Lines 451 - 465
The comparison between aggregated and single-sample co-expression networks, as illustrated by the Jaccard index in Figure 5, provides important insights into the topological coherence of these network types across normal and breast cancer subtypes. The lower similarity observed in normal tissue between its aggregated and individual networks suggests a higher degree of variability in gene co-expression patterns. This variability could be indicative of a broader range of regulatory mechanisms that maintain homeostasis in healthy tissue, as seen in studies where normal networks often exhibit complex dynamic regulatory landscapes[6,67]. In contrast, the higher Jaccard indices in cancer subtypes reflect a greater alignment between aggregated and single-sample networks, suggesting that these subtypes share more common co-expression modules. This increased coherence in cancer may point towards more defined oncogenic programs and conserved regulatory disruptions that support tumor growth and survival. However, the observed decline in similarity among individual networks underscores the inherent heterogeneity within tumor samples, reflecting variations in gene expression and regulatory dynamics that may arise due to genetic mutations, epigenetic alterations, or microenvironmental influences[68,69]. This heterogeneity aligns with the recognized genetic diversity and adaptive capabilities of cancer cells, which are known to exploit flexible co-expression patterns to evade therapeutic interventions. While aggregated networks capture the core regulatory frameworks, single-sample networks reveal unique co-expression patterns that contribute to the personalized complexity of each tumor, highlighting patient-specific genomic interactions that could inform individualized therapeutic strategies[38].
Regarding the survival analysis based on network metrics:
Línea 466 - 487
The results of our survival analysis indicate that the proportion of CIS interactions in breast cancer patients is not significantly related to the overall survival or 5-year survival in molecular subtypes. These results question the idea that intrachromosomal interactions, as represented by the proportion of CIS interactions can play a decisive role in determining the outcome of the patient (Fig 6). Our data do not reveal significant differences in survival between high and low CIS groups in all subtypes. This lack of connection may indicate that genomic stability, particularly in terms of intra-chromosome connectivity, is not a factor in the prediction of breast cancer patients. The slight trend observed in the Basal subtype, although not statistically significant, suggests a weak association that requires further investigation. Here, the concept of chromosomal instability (CIN) and its role in cancer progression provides a possible context for our findings. CIN, characterized by an increased rate of chromosomal gains and losses, is a hallmark of many cancers and contributes to tumor heterogeneity and evolution. It has been observed that while intermediate levels of CIN can promote tumorigenesis by increasing tumor heterogeneity, very high levels of CIN may impair tumor growth due to the generation of nonviable karyotypes[70,71]. Our results may be consistent with these observations, as high levels of CIS interactions, reflecting potential genomic stability, are not associated with better survival outcomes. This suggests that, rather than direct the proportion of CIS, the broader context of genomic alterations and their interactions, including the balance between genomic stability and instability, may play a more nuanced role in breast cancer progression and patient survival. The presence of stable CIS interactions, while reflecting a less chaotic chromosomal environment, may fail to capture the broader genomic instability that drives cancer progression through other mechanisms, such as translocations or structural rearrangements in key oncogenic regions. Additionally, while aneuploidy and CIN are common in many cancers, including breast cancer, their precise impact on survival remains complex and context-dependent[72,73], requiring a more integrative analysis of genomic and clinical data to fully understand their contributions to patient prognosis.
- While the identification of high-degree genes and critical cytobands is valuable, the manuscript would benefit from a deeper biological interpretation. Discuss the known functions of these genes and how they may contribute to breast cancer development or progression. Linking your findings to existing literature can provide context and underscore their significance.
According to the Reviewer’s suggestion, we have increased the discussion regarding the high degree genes in all the phenotypes:
Lines 488 - 508
The analysis of high-degree genes and their cytoband localization in single-sample networks across breast cancer subtypes provides significant insights into the molecular heterogeneity and regulatory disruptions inherent to each subtype. The high recurrence of genes like C1QBP and RPS3A in Luminal A, USP31 and RRM1 in Luminal B, and PSMD8 and YWHAB in Basal subtypes highlights the subtype-specific regulatory networks that underlie tumor progression [74–78]. These high-degree genes are pivotal in maintaining the integrity of the co-expression network, as their centrality in the largest network components indicates their crucial role in regulating multiple pathways. C1QBP has been shown to play a role in cell metabolism and immune evasion, which are critical factors in cancer cell survival and proliferation. Elevated C1QBP expression has been linked to poor prognosis in several cancers, due to its role in promoting cell proliferation and metastasis by modulating mitochondrial function [74]. RPS3A is a component of the ribosomal machinery, and its overexpression is associated with increased protein synthesis, a hallmark of rapidly dividing cancer cells. RPS has been implicated in oncogenic transformation and may contribute to the aggressive growth observed in Luminal A breast cancer by promoting cell cycle progression [75]. In Luminal B, USP31 is involved in immune regulation, particularly in the modulation of the NF-κB pathway, which is frequently dysregulated in cancer. By stabilizing key components of this pathway, USP31 may enhance immune evasion and contribute to the tumor's ability to escape immune surveillance [79]. RRM1 plays a crucial role in DNA synthesis and repair, making it essential for the high proliferative activity of cancer cells. Overexpression of RRM1 has been linked to chemoresistance in breast cancer, as it supports DNA replication in rapidly dividing tumor cells and promotes survival under genotoxic stress [76]. Also, in Basal subtype, PSMD8 is involved in protein degradation via the ubiquitin-proteasome system, which is crucial for maintaining protein homeostasis in cancer cells. Dysregulation of the proteasome system can lead to the accumulation of misfolded proteins, promoting tumor progression and metastasis [77]. YME1L1, a key mitochondrial protease, plays a critical role in regulating mitochondrial dynamics, including the processes of fusion and fission, which are essential for maintaining cellular energy and metabolic homeostasis. Overexpression of YME1L1 in Basal breast cancer could contribute to the aggressive phenotype observed in this subtype by enhancing mitochondrial biogenesis and respiratory function. Given that disruptions in YME1L1 have been linked to impaired mitochondrial structure and increased production of reactive oxygen species (ROS), its upregulation in Basal tumors may support the high metabolic demands and proliferative capacity of these cancer cells.
The cytoband localization of these high-degree genes further underscores their significance in breast cancer biology. Regions such as 11q13.1 and 17q12.1 in Luminal subtypes are known to harbor oncogenes like CCND1, which drives cell cycle progression and is frequently amplified in breast cancer [42–44,80]. The amplification of 11q13.1 has been linked to increased tumor proliferation and worse prognosis in patients. Similarly, the frequent localization of high-degree genes in 8q24.3, a region known for MYC amplification, suggests that this hotspot plays a pivotal role in the aggressive behavior of Luminal B and Her2 subtypes [81,82]. MYC is a master regulator of cell growth and metabolism, and its dysregulation is a hallmark of many cancers, including breast cancer. Its amplification in these subtypes could drive rapid tumor growth and contribute to poor clinical outcomes. For the Basal subtype, frequent cytoband localization in 6p21.1 and 19q13.2 correlates with regions involved in immune modulation and cellular adhesion, crucial processes in the metastatic potential of this aggressive subtype [83,84]. These findings emphasize the importance of cytobands such as 11q13.1, 8q24.3, and 17q12.1 as recurrent regions across multiple subtypes, reflecting shared vulnerabilities in the genomic landscape of breast cancer. Yet, the distinct genetic contexts of each subtype underscore the complexity of their molecular heterogeneity, as evidenced by the differential distribution of high-degree genes. This variation in genomic disruptions and the localization of high-degree genes within key cytobands highlights their functional roles in driving subtype-specific oncogenic processes and potentially influencing treatment resistance.
- Highlight how the identified subtype-specific genes and network disruptions could inform potential therapeutic strategies or biomarkers for personalized medicine.
We have incorporated the Reviewer’s comment into the discussion section, previous to the conclusions.
The identification of patient-specific co-expression networks through single-sample analyses represents a pivotal strategy for advancing precision medicine. By capturing the unique genomic profiles of each patient, this approach reveals the specific regulatory disruptions driving tumor progression in different breast cancer subtypes. Unlike aggregated analyses, which may obscure individual variability, single-sample networks highlight the distinct vulnerabilities of each tumor, enabling a more tailored understanding of the molecular mechanisms at play. In precision oncology, this methodology allows for the identification of key network nodes that could serve as actionable therapeutic targets. By focusing on the unique structural disruptions within an individual’s network, clinicians can prioritize interventions that target these critical hubs, increasing the likelihood of therapeutic efficacy. For example, network hubs with higher connectivity or genomic instability could be targeted with drugs that specifically modulate those pathways, offering a more precise intervention strategy[14].
Moreover, these individualized networks provide insights into potential biomarkers for predicting treatment response or resistance. Patients whose networks exhibit specific disruptions—such as increased intrachromosomal interactions or fragmentation—could be stratified for targeted therapies designed to address those specific genomic irregularities. By leveraging this personalized approach, precision medicine can optimize treatment decisions, reducing unnecessary treatments and improving outcomes by tailoring therapies to the unique molecular landscape of each patient’s tumor. Thus, integrating single-sample network analysis into clinical practice offers a promising path toward more effective and personalized cancer treatment strategies[14].
- While the English language quality is acceptable, there are minor grammatical errors and awkward phrasings that could be improved. Careful proofreading or assistance from a professional editor would enhance readability.
We received the assistance of a professional English Language Editor. Thank you for pointing this out.
Reviewer 2 Report
Comments and Suggestions for Authors
The authors provide an interesting idea of inter and intra chromosomal interactions based on gene regulatory networks. They are further looking into cytoband interactions in breast cancer data. However, the interactions changes from normal to healthy tissue does not provide any significant survival advantages. The authors provide GitHub repository for the code which is commendable. The primary question is whether grns they derived are possibly transcription factor-based network or not. If true, the interactions are based on common transcription factor targets. Since chromosomal interactions are usually meant to be physical such as determined by HiC, are the interactions proposed here similar in nature? do the author propose any different mechanism? Did they observe any concordance with mcf7/10a data as they saw previously. Was the TCGA subtype data derived using the PAM50? Or immunohistochemistry? Have the authors considered the pietenpol subtypes? Another interesting question to ask is to run their interactions without classifying breast cancer by subtypes and then see if classifications overlap with PAM50 or not. The main concern is the lack of significant survival benefits based on their classifications. Is it possible to check cis interactions with genes that are associated with survival and computing interactions between them? It seems the authors only looked at protein coding genes. How will miRNA or other lncRNAs affect their calculations and thereby survival associations? Additionally, it will be better to define the interactions and how interactions were calculated by adding the details in the method section. Please also include how high or low CIS was determined. It is also not very clear how lioness impacted the result. If single sample analyses correlated positively with the overall analysis, where did the result differ? Will it be more meaningful to look into targetable alternation when using single sample analysis? The authors looked into tumor purity, how does that affect their analysis? Did they filter out tumor samples below certain purity estimates?
Author Response
Dear Reviewer,
We would like to extend our sincere thanks for your thoughtful and constructive feedback on our manuscript. We have carefully considered each of your suggestions and have made the necessary revisions to improve the quality of our work in accordance with their recommendations. A revised version of the manuscript, as well as a tracked changes version are attached for your review.
We are confident that these revisions address all the valuable insights provided by both reviewers and significantly enhance the manuscript’s overall quality. We believe the work now offers a substantial contribution to the field of breast cancer genomics. Below, we provide a point-by-point response to your comments. For clarity, our responses are highlighted in bold.
Thank you once again for your time and consideration.
Reviewer 2:
The authors provide an interesting idea of inter and intra-chromosomal interactions based on gene regulatory networks. They are further looking into cytoband interactions in breast cancer data. However, the interaction changes from normal to healthy tissue do not provide any significant survival advantages. The authors provide a GitHub repository for the code, which is commendable.
- The primary question is whether gns they derived are possibly transcription factor-based network or not. If true, the interactions are based on common transcription factor targets.
Thank you for your comment. Our gene coexpression networks are derived from RNA-Seq data and are mainly focused on capturing interactions based on mutual expression patterns between breast cancer subtypes. Although transcription factors (TFs) inherently contribute to coexpression patterns, our networks were not explicitly constructed from transcription factors, but represent broader co-regulatory interactions. In particular, the presence of high-degree genes within critical cytobands in each subtype indicates possible key regulatory nodes, some of which may be transcription factors or their targets. Future work by the group would propose that idea to evaluate the proportion of TF-mediated interactions within these networks and their implications.
- Since chromosomal interactions are usually meant to be physical such as determined by HiC, are the interactions proposed here similar in nature? do the author propose any different mechanism? Did they observe any concordance with mcf7/10a data as they saw previously.
The chromosomal interactions identified in our study, while distinct from physical interactions observed in Hi-C experiments, reveal a transition in breast cancer subtypes from interchromosomal (TRANS) to intrachromosomal (CIS) co-expression patterns. This shift indicates a potential disruption in long-range genomic communication, favoring more localized regulatory interactions. While these co-expression interactions may not directly correspond to physical DNA contacts, they reflect meaningful changes in the underlying genomic architecture.
In this study, we did not perform direct comparisons with MCF7/10A data; however, previous studies (Espinal-Enríquez et al., 2017, RNA-Seq based genome-wide analysis reveals loss of inter-chromosomal regulation in breast cancer) have shown a consistent loss of interchromosomal interactions in breast cancer when comparing co-expression patterns derived from RNA-Seq data to Hi-C-derived interactions in MCF7 and MCF10A cell lines. These findings support the broader trend we observe here in breast cancer subtypes.
It is essential to note, however, that Hi-C experiments capture direct physical interactions between DNA segments, whereas the mutual-information-based co-expression networks in our study represent statistical dependencies among mRNA molecules. Thus, while consistent trends emerge, caution is warranted when interpreting these interactions as they reflect different types of genomic relationships.
- Was the TCGA subtype data derived using the PAM50? Or immunohistochemistry? Have the authors considered the pietenpol subtypes?
The molecular subtype data used in our study, which includes Luminal A, Luminal B, Her2, and Basal breast cancers, were obtained from the TCGA dataset based on RNA-Seq gene expression profiles. These subtypes were classified using the PAM50 intrinsic subtype predictor, as this method is widely used for genomic classification in breast cancer research due to its accuracy in distinguishing molecular subtypes based on gene expression patterns. While the Pietenpol subtypes provide a valuable framework for triple-negative breast cancer (TNBC), our analysis primarily focused on PAM50-defined subtypes to ensure consistency with previous studies in this dataset. Future analyses could consider the Pietenpol classification, particularly for more detailed analyses of TNBC.
- Another interesting question to ask is to run their interactions without classifying breast cancer by subtypes and then see if classifications overlap with PAM50 or not.
We appreciate the reviewer’s suggestion to analyze interactions without classifying breast cancer samples by subtypes. This raises an important question in network biology: whether an expression-derived network can improve the classification of a given set of samples. While we agree this is a valuable and timely problem, which we are currently investigating in other datasets, we consider that a classification analysis lies beyond the scope and objectives of this manuscript. Nonetheless, our preliminary findings support the hypothesis that network structure and metrics may indeed enhance the classification of unclassified samples.
- The main concern is the lack of significant survival benefits based on their classifications. Is it possible to check cis interactions with genes that are associated with survival and computing interactions between them?
In response to the reviewer’s concern regarding the lack of significant survival benefits based on our classifications, we recognize the value of examining cis-interactions specifically among survival-associated genes. By calculating co-expression interactions between these survival-related genes within our single-sample networks, we could investigate whether the presence or absence of specific cis-interactions adds nuance to their prognostic value. Such an approach could indeed illuminate the role of localized genomic interactions in influencing clinical outcomes.
However, it is important to note that this manuscript aims to provide a broad overview of how single-sample co-expression networks can reveal patterns not readily captured by expression data alone, rather than focusing on specific gene-gene interactions or survival-linked nodes. The proposed targeted analysis will therefore be more suitable as part of future research, specifically aimed at identifying gene-specific interactions and their clinical significance. We appreciate the reviewer’s suggestion and believe it will complement the current findings in a follow-up study.
- It seems the authors only looked at protein coding genes. How will miRNA or other lncRNAs affect their calculations and thereby survival associations?
In our study, we focused on protein-coding genes due to their well-documented roles in gene regulatory networks and established associations with breast cancer phenotypes. We acknowledge that miRNAs and lncRNAs are critical gene expression regulators, functioning at post-transcriptional and epigenetic levels, and their inclusion could introduce additional regulatory layers that might influence co-expression patterns and survival associations.
We have previously published studies examining the impact of miRNAs and lncRNAs in cancer contexts, such as breast cancer (Drago-García et al., 2017, Sci Rep), breast cancer molecular subtypes (de Anda-Jáuregui et al., 2019, Appl Network Sci), renal carcinoma (Zamora-Fuentes et al., 2020, Front Genet), and hematopoietic cancers (Nakamura-García & Espinal-Enríquez, 2023, Sci Rep). For instance, miRNAs target multiple mRNAs, which may reshape network architecture by influencing hub genes or altering high-degree nodes, while lncRNAs can scaffold chromatin modifiers or interact with proteins and RNAs, affecting co-expression dynamics.
Although these additional regulatory layers could further refine network structure and survival associations, our current analysis, focused on protein-coding genes, remains robust and biologically relevant. Future studies incorporating miRNA and lncRNA data could provide a more comprehensive view of breast cancer regulatory networks, capturing additional tumor heterogeneity and potentially enhancing prognostic insights.
- Additionally, it will be better to define the interactions and how interactions were calculated by adding the details in the method section.
According to the reviewer’s suggestion, we have modified the methods section in relation to the definition of a gene-gene interaction.
The revised version is written here:
“We implemented the ARACNe algorithm[22] to infer GCNs for our data, which calculates the mutual information (MI) between gene expression profiles to identify significant coexpression connections. Specifically, ARACNe detects nonlinear relationships between gene pairs by estimating their mutual information, ensuring that only the strongest direct gene-gene interactions are retained. A multi-core version of ARACNe [23] was used to enhance computational efficiency, leveraging parallel processing across multiple cores. For each subtype and normal tissue, the GCN was computed for approximately 12.5 million possible interactions between 5,000 genes (https://github.com/josemaz/aracne-multicore). To ensure comparability across networks, we selected the top 10,000 gene interactions for further analysis.”
“We inferred individual co-expression networks for each breast cancer sample using the LIONESS equation [12]. This approach iteratively excludes one sample at a time from the full cohort, constructing GCNs both with and without the excluded sample. The difference between these two networks reveals the unique co-expression structure for each sample. Then, we calculated MI, to quantify the statistical dependencies between gene pairs, forming the foundational connections of these networks. To enhance computational efficiency, we integrated LIONESS with a parallelized ARACNe implementation, allowing MI to be calculated across multiple cores, thus inferring each individual network in a sequential manner. This implementation can be found at https://github.com/PatricioLOPSA/LIONESS-MI.git. This method provides a sample-specific view of gene coexpression patterns, reflecting individualized genomic interactions. Each single-sample network was subsequently analyzed by focusing on the top 10,000 edges, ranked by their LIONESS-MI scores, which highlighted the most significant coexpression interactions, to ensure consistency and comparability across networks.”
- Please also include how high or low CIS was determined.
The corresponding section in which we define the high and low cis proportion is described here:
“The survival analysis was conducted using Kaplan-Meier estimations to compare overall survival between median-high and median-low CIS (intrachromosomal interaction) groups across different breast cancer subtypes. The CIS proportion for each sample was calculated as the ratio of intrachromosomal interactions to the total number of co-expression interactions within each single-sample network. To classify samples into high and low CIS groups, we used the median CIS value as a cutoff. Samples with a CIS proportion above the median were categorized as high CIS, while those below the median were assigned to the low CIS group. Clinical data were merged with CIS group information, and survival outcomes were assessed for both overall survival and a 5-year survival endpoint (1,825 days). Survival curves were generated using the Kaplan-Meier method, with statistical significance evaluated by the log-rank test. Additionally, Cox proportional hazard models were employed to estimate hazard ratios between high and low CIS groups. Visualizations included confidence intervals and risk tables for each group, enhancing the interpretability of differences in survival probability. The results were plotted using the “ggsurvplot” function from the survminer R package.”
- It is also not very clear how lioness impacted the result. If single sample analyses correlated positively with the overall analysis, where did the result differ? Will it be more meaningful to look into targetable alternation when using single sample analysis?
The integration of LIONESS into our analysis added a crucial layer of granularity by allowing us to construct individualized co-expression networks for each patient, capturing unique genomic interaction patterns that are otherwise obscured in aggregated analyses. While our single-sample networks did correlate with general trends observed in the aggregated network—such as the transition from interchromosomal (TRANS) to intrachromosomal (CIS) interactions—LIONESS uncovered additional nuances, particularly in genomic fragmentation and the localized organization of co-expression patterns within each tumor. This more detailed view highlighted a greater heterogeneity in the degree of disruption across tumors, revealing patient-specific variations that were masked in the aggregated approach. For example, some single-sample networks exhibited markedly higher levels of localized co-expression disruptions, indicating distinct regulatory reconfigurations in individual tumors.
This patient-level heterogeneity implies that key alterations, such as high-grade genes or localized chromosomal instabilities, may not be uniformly present across all patients of a given subtype. These findings suggest the presence of subtype-specific, yet patient-unique dysregulations, which could serve as a basis for personalized therapeutic strategies. By revealing individual deviations in co-expression patterns, LIONESS allows us to identify potentially actionable genomic alterations that would be overlooked in aggregated networks. Such alterations could represent targetable vulnerabilities, especially where high-grade genes or specific chromosomal regions show consistent dysregulation in individual patients. Consequently, LIONESS not only enhances our understanding of the molecular complexity of breast cancer but also offers a pathway to link patient-specific network alterations to potential therapeutic interventions, supporting a more precise approach to cancer treatment. We have now included a more detailed discussion on the impact of LIONESS in analyzing breast cancer genomic data.
“Importantly, the individualized nature of this analysis resolves many of the limitations associated with aggregated network studies, where data from multiple samples are combined, potentially masking the unique genomic interactions that occur within individual tumors. In this regard, single-sample networks offer a more precise view of tumor-specific genomic coordination and the localized genomic disruptions characteristic of different types of cancer. This individual-based approach is crucial, as genomic heterogeneity is a well-established feature of cancer, and aggregated analyses often fail to capture the full extent of variability between tumors. Studies of individual genomic profiles have demonstrated that single-sample analyses are more effective at identifying clinically relevant mutations and genomic alterations that drive tumor behavior, providing a clearer path for personalized therapeutic interventions. The increase in CIS interactions and the reconfiguration of co-expression networks toward intra- and inter cytoband interactions supports the notion of increased genomic instability. These changes align with prior research showing that cancer cells often reorganize their chromatin structures to enhance localized genomic interactions, promoting the formation of transcriptional hubs that regulate oncogene expression and contribute to tumor growth. Furthermore, capturing heterogeneity in coexpression patterns at the individual level would enable the identification of unique genomic alterations that could foster the development of tailored therapeutic strategies. Thus, integrating genomic analyses of individual samples into oncology research may provide deeper insight into underlying genomic instability and help identify potential biomarkers or novel patient-specific therapeutic targets.”
“The identification of patient-specific co-expression networks through single-sample analyses represents a pivotal strategy for advancing precision medicine. By capturing the unique genomic profiles of each patient, this approach reveals the specific regulatory disruptions driving tumor progression in different breast cancer subtypes. Unlike aggregated analyses, which may obscure individual variability, single-sample networks highlight the distinct vulnerabilities of each tumor, enabling a more tailored understanding of the molecular mechanisms at play. In precision oncology, this methodology allows for the identification of key network nodes that could serve as actionable therapeutic targets. By focusing on the unique structural disruptions within an individual’s network, clinicians can prioritize interventions that target these critical hubs, increasing the likelihood of therapeutic efficacy. For example, network hubs with higher connectivity or genomic instability could be targeted with drugs that specifically modulate those pathways, offering a more precise intervention strategy.
Moreover, these individualized networks provide insights into potential biomarkers for predicting treatment response or resistance. Patients whose networks exhibit specific disruptions—such as increased intrachromosomal interactions or fragmentation—could be stratified for targeted therapies designed to address those specific genomic irregularities. By leveraging this personalized approach, precision medicine can optimize treatment decisions, reducing unnecessary treatments and improving outcomes by tailoring therapies to the unique molecular landscape of each patient’s tumor. Thus, integrating single-sample network analysis into clinical practice offers a promising path toward more effective and personalized cancer treatment strategies “
- The authors looked into tumor purity, how does that affect their analysis? Did they filter out tumor samples below certain purity estimates?
In our study, the impact of tumor purity on coexpression networks was carefully considered, as variations in purity could introduce noise or confounding factors into the analysis. However, we did not impose an explicit filtering threshold based on tumor purity estimates. Instead, we used robust normalization and noise reduction techniques (e.g., ARSyN) to mitigate potential biases from mixed cell populations. This approach allowed us to maintain the integrity of our coexpression networks while accounting for the inherent heterogeneity of tumor samples, without excluding any based on purity thresholds alone. Thank you for pointing this out.